# Oral Administration of a Phage Cocktail to Reduce *Salmonella* Colonization in Broiler Gastrointestinal Tract—A Pilot Study

**DOI:** 10.3390/ani12223087

**Published:** 2022-11-09

**Authors:** Wattana Pelyuntha, Ananya Yafa, Ruttayaporn Ngasaman, Mingkwan Yingkajorn, Kridda Chukiatsiri, Nidanut Champoochana, Kitiya Vongkamjan

**Affiliations:** 1Department of Biotechnology, Faculty of Agro-Industry, Kasetsart Univerisity, Chatuchak, Bangkok 10900, Thailand; 2Faculty of Veterinary Science, Prince of Songkla University, Hat Yai 90110, Thailand; 3Department of Pathology, Faculty of Medicine, Prince of Songkla University, Hat Yai 90110, Thailand; 4Faculty of Animal Sciences and Technology, Maejo University, Nongharn, Sansai, Chiang Mai 50290, Thailand; 5Division of Health and Applied Sciences, Faculty of Science, Prince of Songkla University, Hat Yai 90110, Thailand

**Keywords:** broiler, biocontrol, food production animals, phage treatment, *Salmonella*, salmonellosis

## Abstract

**Simple Summary:**

*Salmonella* contamination in the poultry supply chain is a global public concern and can lead to a serious foodborne illness; thus, effective control measures are needed. Here, we focused on the application of phage treatment to control *Salmonella* during the pre-harvest step of poultry meat production. *Salmonella* reduction was monitored in broilers on a commercial farm. Phage treatment has been proposed as a potential alternative biocontrol agent for agricultural purposes due to its specificity and safety, whereby it will only target bacterial hosts without harmful effects on human and animal health. Phages not only provide benefits for reducing the number of pathogens, but they are also easy to apply or deliver to a wide range of animals. In this study, phages and a phage cocktail were developed to provide an effective method for reducing *Salmonella* colonization in the gastrointestinal tract of broilers to improve food safety from farm to fork.

**Abstract:**

*Salmonella* contamination in poultry meat products can lead to serious foodborne illness and economic loss from product recalls. It is crucial to control *Salmonella* contamination in poultry from farm to fork. Bacteriophages (phages) are viruses of bacteria that offer several advantages, especially their specificity to target bacteria. In our study, three *Salmonella* phages (vB_SenS_KP001, vB_SenS_KP005, and vB_SenS_WP110) recovered from a broiler farm and wastewater treatment stations showed high lysis ability ranging from 85.7 to 96.4% on over 56 serovars of *Salmonella* derived from several sources, including livestock and a broiler farm environment. A three-phage cocktail reduced *S.* Enteritidis and *S.* Typhimurium, in vitro by 3.9 ± 0.0 and 3.9 ± 0.2 log units at a multiplicity of infection (MOI) of 10^3^ and 3.8 ± 0.4 and 4.1 ± 0.2 log units at MOI of 10^4^ after 6 h post-phage treatment. A developed phage cocktail did not cause phage resistance in *Salmonella* during phage treatments for three passages. Phages could survive under simulated chicken gastrointestinal conditions in the presence of gastric acid for 2 h (100.0 ± 0.0% survivability), bile salt for 1 h (98.1 ± 1.0% survivability), and intestinal fluid for 4 h (100 ± 0.0% survivability). Each phage was in the phage cocktail at a concentration of up to 9.0 log PFU/mL. These did not cause any cytotoxicity to human fibroblast cells or Caco-2 cells as indicated by the percent of cell viability, which remained nearly 100% as compared with the control during 72 h of co-culture. The phage cocktail was given to broilers raised in commercial conditions at a 9 log PFU/dose for five doses, while naturally occurring *Salmonella* cells colonized in the gastrointestinal tract of broilers were significantly reduced as suggested by a considerably lower *Salmonella* prevalence from over 70 to 0% prevalence after four days of phage treatment. Our findings suggest that a phage cocktail is an effective biocontrol agent to reduce *Salmonella* present in the guts of broilers, which can be applied to improve food safety in broiler production.

## 1. Introduction

*Salmonella* is a major cause of foodborne disease, salmonellosis, due to the consumption of contaminated food, especially poultry meat products [1,2]. In the European Union (EU), salmonellosis is the second most frequently reported zoonotic disease in humans. In 2019, approximately 88,000 salmonellosis cases in EU member states were reported. *Salmonella enterica* serovars, Enteritidis and Typhimurium, are the most common causes of disease outbreaks [3]. *Salmonella* in broilers is commonly reported, with the prevalence found in different regions throughout the world ranging from 1.57 to 36.20% [4,5,6,7,8]. *Salmonella* typically resides inside the gastrointestinal (GI) tract of broilers and can later contaminate poultry meat products during the processing step in the slaughterhouse [9].

Control of *Salmonella* in primary poultry production or at the farm level is immediately required to prevent the introduction of this pathogen into the food chain to, ultimately, reduce foodborne illness cases [10,11]. Previous studies have used various interventions to mitigate *Salmonella* colonized in the broiler production chain, including antimicrobials, vaccination, phytochemicals, and/or probiotics [12,13,14]. Feed additives such as antimicrobials are also common, which can prevent chronic infection in broilers [15,16]. The overuse of antimicrobials, especially antibiotics, leads to the spread of bacterial resistance in a broiler farm environment and the subsequent food chain [15,17]. The spread of antibiotic resistance may cause serious infections and the failure of treatment in humans [18,19,20,21].

Bacteriophages (phages) have been studied as an alternative to antibiotics for agriculture and medical purposes to target specific pathogenic bacteria. Phages have been granted GRAS (Generally Recognized as Safe) status by the USA FDA to be used as antimicrobial food additives. They are capable of reducing the load of bacteria in food products of animal origin [22]. Phage applications have been reported to be safe for use in livestock and poultry [23,24], as they do not infect animal cells but are rather specific to only bacterial receptors [25,26]. Phages targeting *Salmonella* in broilers have been successfully examined in several trials [10,27]. A previous study reported that phage-treated broilers showed lower colonization of *S.* Enteritidis in caecal content when compared to the control group [28]. SalmoFree®, a commercial phage cocktail, could reduce *Salmonella* present in a large-scale farm (>34,000 birds) as *Salmonella* incidence in cloacal samples in the phage-treated houses dropped to an undetectable level when compared to control houses at the end of a 34-day study under commercial rearing conditions [10].

In the current study, phage cocktail application for broilers was explored to evaluate the efficacy against *Salmonella* colonization in broiler gastrointestinal tracts. *Salmonella* phages in our collection were previously isolated from natural sources and were used to prepare a phage cocktail for treating broilers on a commercial farm. This study was conducted as a preliminary study with broilers under commercial rearing conditions to obtain basic information for further design of the phage treatment program to be applied to broilers at a larger scale. Our study also investigated phage-resistance development in *Salmonella*, phage survivability under simulated chicken gastrointestinal conditions, and the phage cytotoxic effect on cell lines to obtain information on the safety of phage application.

## 2. Materials and Methods

### 2.1. Bacterial Strains Used and Culture Conditions

Fifty-six serovars of *Salmonella* in our collection were used in this study (Appendix A). These serovars were previously isolated from several sources, including commercial broiler farms, free-range farms, slaughterhouses, and stalls from wet markets [22,29,30,31]. Additional serovars from the culture collection of the Department of Medical Sciences, Ministry of Public Health, Thailand, were also included. These bacterial serovars were propagated in tryptic soy broth (TSB) (HiMedia Laboratories, India) at 37 °C for 18 to 24 h prior to the study. All bacterial strains were maintained in 15% (*v*/*v*) glycerol solution and kept under −20 °C at the Department of Biotechnology, Faculty of Agro-Industry, Kasetsart University, Chatuchak, Bangkok, Thailand.

### 2.2. Phage Lysate Preparation, Host Range Determination, and Efficiency of Plating

Three phages, including vB_SenS_KP001 (KP001), vB_SenS_KP005 (KP005), and vB_SenS_WP110 (WP110) in our collection, were selected for this study. Phages were originally recovered from sewage from a broiler farm (KP001 and KP005) and wastewater from a wastewater treatment station (WP110). *S.* Anatum A4-525 was used as a natural host of KP001 and KP005, whereas *S.* Kentucky S1H28 was the host for phage WP110 [8,29]. A double-layer agar assay and phage suspension harvesting were performed using salt magnesium (SM) buffer. Filtrates were collected after being passed through 0.20 µm syringe filters and stored at 4 °C until analysis [8,29,31]. Phage titers were enumerated by observing plaques presented on each plate of given dilutions. Phage host range determination was performed on the culture lawn of each *Salmonella* serotype according to a previously published protocol [8,31]. *Escherichia coli* ATCC 25,922 was used as a standard control. All plates were observed for the appearance of clear plaques after growth at 37 °C for 18 h. The efficiency of plating (EOP) was determined on two representative strains (*S.* Enteritidis S5-371 and *S.* Typhimurium S5-370). The EOP was calculated using the given Formula (1):EOP = average PFU on target bacteria/average PFU on host bacteria(1)

EOP was classified as “high production” efficiency when the ratio was 0.5 or more. An EOP of 0.1 or higher but below 0.5 was considered “medium production” efficiency, and that between 0.001 and 0.1 was considered “low production” efficiency. An EOP of 0.001 or below, and when any dilution did not result in any plaque formation, were classified as inefficient [8].

### 2.3. Phage Cocktail Development and Phage Efficacy Test In Vitro

Each phage was mixed in an equivalent quantity to obtain a working cocktail stock. A developed phage cocktail was assessed for its efficacy on *S.* Enteritidis S5-371 and *S.* Typhimurium S5-370. A bacterial strain suspension at the final concentration of 4 log CFU/mL was combined with a phage cocktail at the final concentration of 7 and 8 log PFU/mL to achieve effective MOIs of 10^3^ (low) and 10^4^ (high), followed by incubation with agitation (220 rpm) at 37 °C for 18 h. In addition, the effectiveness of a single phage used in a cocktail was evaluated on *S.* Enteritidis S5-371 and *S.* Typhimurium S5-370 to compare to the efficacy of a phage cocktail on the same strains. A *Salmonella* suspension with an initial 4 log CFU/mL was mixed with a single phage at a concentration of 7 log PFU/mL (MOI 10^3^). The culture of each *Salmonella* strain without single phages or a phage cocktail was kept as a control. A number of *Salmonella* cells in the control and treatment conditions were evaluated every 6 h using a spread plate technique on TSA plates [8].

### 2.4. Study of Phage-Resistance Development in Salmonella upon Phage Cocktail Treatment

The change of resistance phenotype of *Salmonella* after treatment with a phage cocktail was investigated by a spot test. A culture of *Salmonella* (4 log CFU/mL) was mixed with a phage cocktail (6 log PFU/mL) and incubated at 37 °C (220 rpm) for 18 h. A loopful of culture was streaked on a TSA plate to obtain a single colony, which was subsequently used to prepare a double-layer plate for a spot test to evaluate phage resistance. The remaining culture in the same tube was used for an additional two passages following previously published protocols [8,29].

### 2.5. Phage Survivability in Simulated Chicken Gastrointestinal Conditions

For survivability in gastric conditions, a phage cocktail at 8 log PFU/mL was added into the stimulated gastric fluid (SGF) comprised of 0.6 g/mL pepsin (Sigma Aldrich, St. Louis, MO, USA) in 0.2% NaCl solution (pH 2.0), followed by incubation at 37 °C for 2 h. The mixture was then transferred to 2% (*w*/*v*) bile salt solution and incubated at 37 °C for 1 h. For survivability in intestinal conditions, the stimulated intestinal fluid (SIF) comprised of 1 g/mL pancreatin (Sigma Aldrich, USA) in 0.2% NaCl solution (pH 8.5) was used to treat the phage cocktail and then incubated at 37 °C for 4 h. The phage titers were determined as mentioned above.

### 2.6. Evaluation of Phage Cytotoxicity in Cell Lines

The cytotoxic effect of phages on skin cells was evaluated by utilizing primary human dermal fibroblast (HDFn) cells PCS-201-010^TM^ from a stock culture seeded in a 6-well microtiter plate (#140675, Nunc™ Cell-Culture Treated Multidishes, Thermo Fisher Scientific, Waltham, MA, USA) and grown in Dulbecco’s Modified Eagle Medium/ Nutrient Mixture F-12 (DMEM/F-12) (Gibco, Thermo Fisher Scientific, Waltham, MA, USA), supplemented with 1% (*v*/*v*) penicillin (10,000 U/mL), 20% (*v*/*v*) fetal bovine serum, and 2 mM L-glutamine under a 5% saturated CO_2_ atmosphere at 37 °C. A representative intestinal cell line, Caco-2 cells HTB-37™, was also used for the evaluation of phage cytotoxicity. Caco-2 cells were seeded in a 96-well microtiter plate (#140675, Nunc™ Cell-Culture Treated Multidishes, Thermo Fisher Scientific, Waltham, MA, USA) and grown in complete Dulbecco’s Modified Eagle Medium (DMEM) (Gibco, Thermo Fisher Scientific, USA), supplemented with 1% (*v*/*v*) penicillin/streptomycin, 10% (*v*/*v*) fetal bovine serum, and 2 mM L-glutamine under a 5% saturated CO_2_ atmosphere at 37 °C and 95% humidity. A single phage (10 µL) with a titer of 6.0 to 9.0 log PFU/mL was co-cultured to fibroblast cells (6 log cell/mL) or Caco-2 cells (4 log cell/mL), followed by incubation under a 5% saturated CO_2_ atmosphere at 37 °C for 72 h. The cell culture without a phage suspension was used as a control. The culture fluid was also collected at every 24 h intervals and investigated using sulforhodamine B colorimetric assay to determine the percentage of fibroblast cell viability, according to the modified protocol of [32]. The culture fluid of Caco-2 cells was treated with 5 mg/mL of MTT reagent (Sigma Chemical CO, St Louis, MO, USA) for 3 h, followed by the addition of DMSO, and the optical densities were measured at 540 nm.

### 2.7. Efficacy of a Phage Cocktail in Reducing Salmonella Populations in a Broiler Gastrointestinal Tract

This study was approved by the Animal Care and Use for Science and Technology from Maejo University under the approval number MACUC013A/2565. This trial was carried out under commercial rearing conditions in a commercial broiler farm containing over 24,000 birds located in Songkhla Province, Thailand. A total of 30 broilers (Ross 308) aged 11 days at a weight of approximately 440–480 g were included in a separate area within a broiler house. Broilers were randomly divided into three groups (ten chickens/group) in separate pens. Chicken feed and water were provided by a pan feeding and nipple waterer system, along with continuous lighting. Broilers in group I were the controls, which were treated with SM buffer by oral administration via a feeding tube. Broilers in groups II and III were the phage-treated groups, which were treated with a phage cocktail (9 log PFU/mL) by oral administration using a feeding tube (1 mL for each dose), and nipple waterer (adding 200 mL of 10 log PFU/mL to 1.8 L of water for 30 min drinking time), respectively. Each dose of phage cocktail and SM buffer was delivered to the broilers on days 0, 4, 13, 16, and 20. In addition, clinical signs, diarrhea, rash and irritation, and certain respiratory or gastrointestinal disorders were observed via visual inspections by a specialized avian veterinarian.

### 2.8. Sampling Method and Sample Examination

Cloacal samples were collected before each dose of phage and was given on day 0 (baseline measurement), 4, 13, 16, and 20 by swabbing broilers with individual cotton sticks and transferred to 9 mL buffered peptone water (BPW) supplemented with *Salmonella* Supp Tab (#421202, Biomérieux, Marcy-l’Étoile, France). Swabs were stored in an icebox (4 °C) during transportation to the laboratory for analysis (~1–2 h). Sampling was carried out first on the control group to prevent phage cross-contamination. Additionally, broilers in each group (*n* = 5) were randomly selected for termination on day 20 of the study. Their entrails, including the liver, caecum, and spleen, were aseptically collected for the detection of invasive *Salmonella* infection in their internal organs.

Cloacal samples were processed for *Salmonella* detection, and individual samples in BPW tubes were incubated at 41.5 °C for 24 h. A loopful of culture fluid was further streaked on an XLD agar or SALMA® agar plate (Biomérieux, Marcy-l’Étoile, France) according to the modified ISO 6579: 2017, provided by the Biomérieux company, Marcy-l’Étoile, France. The typical red colonies with a black center in the XLD plate or purple colonies on the SALMA® agar were observed and recorded. Biochemical tests, including urease, lysine iron sugar (LIA), and triple sugar iron (TSI), were used for *Salmonella* confirmation according to the ISO 6579: 2017 recommendation. Suspected isolates of *Salmonella* were further confirmed via latex agglutination test by a commercial service company (S&A Reagents Lab. Ltd., Part., Bangkok, Thailand). Entrail samples, including the liver, caecum, and spleen, were aseptically cut into small pieces (5–10 g) and mixed with BPW at a ratio of 1:9 in a stomacher bag and vigorously blended with a circulator lab blender (Seward Stomacher® Model 400, Seward Ltd., Worthing, West Sussex, U.K.). The mixture was further incubated at 37 °C for 24 h and processed as per the protocol mentioned above. The prevalence of *Salmonella* in all samples was reported.

### 2.9. Statistical Analysis

Statistical analysis was performed using SPSS (version 22.0) of Windows statistics software (SPSS Inc., Chicago, IL, USA). Data of *Salmonella* cell reduction for each period were subjected to an analysis of variance, followed by Tukey’s range test. A significant difference between the control and the phage treatments for the phage efficacy at different MOIs and the phage cytotoxicity was calculated using the independent samples *t*-test. For the phage cytotoxicity, statistical analysis was performed only to the percent cell survivability of < 100% due to cell proliferation allowing the percent cell survivability to be >100%. A difference was considered statistically significant at a *p* < 0.05.

## 3. Results

### 3.1. Phage Host Range against Diverse Salmonella Serovars and Efficiency of Plating

Phages KP001 and KP005 showed broad lysis ability against 48 of 56 serovars (85.7%) of *Salmonella* included in this study. Both phages could not lyse *S.* Agona, *S.* Albany, *S.* Dublin, *S.* Hadar, *S.* Choleraesuis, *S.* Paratyphi A, and *S.* Typhi. In addition, phage KP001 could not lyse *S.* Cerro, whereas vB_SenS_KP005 could not lyse *S. Gallinarum*. Overall, phage WP110 showed the strongest lytic ability as it could lyse most serovars tested (96.4%). However, *S.* Mbandaka and *S.* Infantis appeared to be resistant to this phage (Figure 1). In addition, three phages could not lyse a standard control *E. coli* ATCC 25,922.

The result of the EOP assay revealed that the KP001 phage displayed a medium production efficiency on both *Salmonella* representative serovars (0.25 ± 0.01 and 0.48 ± 0.03). This was the same as phage KP005, which also showed the same medium production efficiency (0.18 ± 0.02 and 0.14 ± 0.03). Phage WP110 showed high production efficiency for *S.* Enteritidis S5-371 (5.20 ± 1.13) and a medium production efficiency for *S.* Typhimurium S5-370 (0.47 ± 0.14). In addition, three phages showed ineffective production efficiency on a standard control *E. coli* ATCC 25,922 (no plaque observed in any dilutions). Hence, the three phages included here were further used for the in vitro and in vivo studies.

### 3.2. Evaluation of a Phage Cocktail Targeting Salmonella In Vitro

A phage cocktail treatment at both MOIs (medium and high) significantly reduced *Salmonella* cells by approximately 4 log CFU/mL, suggesting a 100% reduction for *S.* Enteritidis and *S.* Typhimurium at 6 h of post-phage treatment (Table 1). *Salmonella* counts were not detected (ND) at 12 and 18 h of post-phage treatment when compared to the control. Treatment with a single phage used in a phage cocktail showed the reduction of *Salmonella* cells by 2–3 log units at 6 h (Table 2). A significant difference was observed when compared with the phage cocktail treatment (Table 1). The phage cocktail completely reduced the population to an undetectable level (ND) at the same time.

### 3.3. Study of Phage-Resistance Development in Salmonella after Treatment with Phage Cocktail

Isolates of *S.* Enteritidis and *S.* Typhimurium after treatment with a phage cocktail were tested with the same phage cocktail for three passages. There was no sign of phage-resistance development in both reference strains when compared with the initial lysis profiles of the control culture (Table 3).

### 3.4. Phage Survivability in Simulated Chicken Gastrointestinal Conditions

The total phage counts in a phage cocktail presented 100% survivability (9.1 ± 0.1 log PFU/mL) in gastric conditions (pH 2.0), up to 98.1 ± 1.0% survivability (9.0 ± 0.2 log PFU/mL) in bile salt conditions, and 100% (9.1 log PFU/mL) in intestinal conditions (pH 8.5) when compared to the phage counts in a control suspension.

### 3.5. Evaluation of Phage Cytotoxicity in Cell Lines

Both human dermal fibroblast and Caco-2 cells tested with each phage maintained high cell viability of nearly 100% for all phage concentrations tested (6.0 to 9.0 log PFU/mL) from 24 to 72 h, as shown in Table 4 and Table 5, respectively.

### 3.6. Reduction of Salmonella in the Broiler Gastrointestinal Tract after Phage Cocktail Treatment

In the baseline measurement prior to a phage treatment (visit 1; day 0), *Salmonella* was detected in samples from group I (control), presenting an 80% prevalence (Table 6). Similar *Salmonella* counts were detected on day 0 in group II and III samples with a prevalence of 90 and 70%, respectively. At visit 2 (day 4), *Salmonella* was not detected in all cloacal samples of the broilers in group II (phage delivery via a feeding tube) (0% prevalence). However, *Salmonella* was still detected in samples from group III (phage delivery via nipple waterer) at a prevalence of 40%. During visits 3 (day 13), 4 (day 16), and 5 (day 20), *Salmonella* was not detected in both phage-treated groups II and III (0% prevalence). Samples from the control group I were still positive for *Salmonella*, presenting a prevalence of 60, 100, and 100% during visits 3 (day 13), 4 (day 16), and 5 (day 20), respectively. The result showed no invasive *Salmonella* detected (ND) in all organ samples (liver, spleen, and caecum) tested. The use of the phage cocktail did not cause any side effects or have any harmful effects on animal health during this study. Neither clinical signs nor mortality was observed in any group during the entire experiment. Moreover, two district serovars, including Typhimurium and Corvallis, were characterized and remained contaminated during assay in this broiler farm.

## 4. Discussion

Three *Salmonella* phages, KP001, KP005, and WP110 from our collection, were selected to investigate their ability to reduce *Salmonella* colonization in broiler intestinal tracts. *Salmonella* phages have been widely recovered from environmental samples, especially sewages and wastewater from animal farms [22,33,34]. In our study, each phage showed a broad and extended host range when combined in a phage cocktail. Bacterial hosts included in the host range determination were originally isolated from various animals and primary production sources. Since most *Salmonella* hosts were susceptible to each phage, which were previously isolated from sewage and wastewater from broiler farms, phage-host ecology may suggest a broad host range characteristic of these phages included in a cocktail [35,36]. In addition, phages may live together with their predominant hosts, and their lysis profile might be linked with the origin of the isolated host [33,37,38]. *Salmonella* phages showed an ability to lyse over 20 serovars, which suggests they are a potential biocontrol agent against *Salmonella* serovars [22,29,35,36]. In this study, three phages showed strong lysis patterns against serovars from different sources, suggesting that their lytic activities are advantageous and preferred in fighting *Salmonella* prevalent in broiler farms. Several studies have revealed that the application of mixed phages, or a phage cocktail, is usually preferred over a single-phage treatment due to its high efficiency in reducing bacterial populations in vitro [39,40,41,42]. A single *Salmonella* phage might cover a specific spectrum of the host serovars, whereas multiple phages can expand their lysis ability and cover more serovars, as we achieved here. Both MOIs (medium and high) of our developed phage cocktail were highly effective, as indicated by a 4 log reduction (100%) in vitro (4 log units) after 6 h of phage treatment initiation. Similar to the findings of other studies, a phage cocktail composed of three *Salmonella* phages could reduce *Salmonella* populations by 4 log units 4 h after phage treatment [29]. Phages in a commercial product, SalmoLyse®, decreased *Salmonella* populations by 90% after being treated with 8 log PFU/mL for 1 h [43]. The same study revealed that this phage cocktail was effective in reducing 95% of tested strains in their library. In our previous work, a phage cocktail of vB_SenS_WP109, vB_SenS_WP110, and vB_SenP_WP128 could reduce the counts of eight *Salmonella* strains derived from broiler farms and five foodborne outbreak-related isolates from 2.2 to 2.8 log units at 6 h of treatment. This phage cocktail did not change *Salmonella* phenotypic characteristics, and no phage resistance was developed upon treatment with the phage cocktail for three passages, as observed in a previous study [8]. Treatment with a single phage may lead to the possibility of phage-resistance development in bacteria. Here, we see the benefits of using a phage cocktail because if bacteria develop a resistance to one phage, they might be susceptible to other phages present in a cocktail. Overall, the use of a phage cocktail is a potential approach for controlling *Salmonella* that has been distributed or contaminated in broilers and a farm environment.

In our study, the toxicity of phages included in a cocktail was examined on human dermal fibroblast and Caco-2 cells. There was no sign of toxic effects upon phage treatment at the level of up to 9.0 log PFU/mL on both cell types. Findings here suggested the safety level for handling and treating phages to broilers. In another study, *Acinetobacter baumannii* phage BS46 at concentrations of 2 × 10^9^ PFU/mL showed no impact on the viability of mouse embryonic fibroblast 3T3 cells after 24 h of exposure [44]. The toxicity of phages was also confirmed by [45]. Two enterococci phages, namely: vB_EfaS-Zip (8 log PFU) and vB_EfaP-Max (7 log PFU), were non-toxic to 3T3 cells by showing 100% cell viability.

Our phage cocktail showed 100% survivability under simulated chicken gastrointestinal conditions. This assumes that our developed phage cocktail is active and survived in each part of the broiler gastrointestinal tract. Overall, phages are exposed to extremely low pH (1.5–3.5) and high concentrations of hydrogen ions presenting within chicken gizzards, with bile salt in the proximal intestine and high pH (8–9) at the cloaca [46]. The applications of a phage cocktail for reducing *Salmonella* have been previously reported by several authors. For example, a phage cocktail composed of UAB_Phi20, UAB_Phi78, and UAB_Phi87 (10 log PFU/animal per dose) could decrease *Salmonella* colonized in the cecum by 1 log unit after five and six days post-challenge and by 0.5 log units after 12 days post-challenge [47]. The same study revealed that pre-treatment with phages, prior to a challenge with *Salmonella* for one day, could decrease the colonization of this bacterium by 4.4 and 3.2 log units on day two and six and 2 log units on day eight. In our study, after a phage cocktail was given to broilers for one dose at 11 days of age, *Salmonella* could be reduced from their gastrointestinal tracts as the prevalence of *Salmonella* in all cloacal samples was reduced from 100 to 40%, and from 100 to 0% by phage treatments via a feeding tube and a nipple waterer, respectively, while the prevalence in the untreated control group remained at 100%.

Overall, the preliminary data presented here suggested that broilers in the farm included in this study showed a high prevalence of *Salmonella* contamination, ranging from 70–90% prevalence in cloaca during days 1–11 of broiler rearing. Oral delivery via a feeding tube seems to be preferable to a nipple waterer because of a complete phage dosage being fully administered and later directly transferred into the gastrointestinal tract of animals without any interruptions. However, phage administration through a nipple waterer may be encountered by the movement of water in the pipeline and inconsistent phage dosing or concentrations that were received by each broiler. Overall, this study suggests that a phage cocktail treatment during broiler rearing under commercial conditions can reduce *Salmonella* populations in broiler guts, which can increase the safety of poultry starting from the farm.

## 5. Conclusions

The application of a phage cocktail as an oral administration through a feeding tube or nipple waterer can reduce *Salmonella* colonization in the gastrointestinal tract of broilers. The phage cocktail did not have side effects on broilers’ growth and health. Our phage cocktail developed here was effective and could be considered as a potential biocontrol in broilers to increase the safety of food of animal origin, especially poultry meat. However, the efficacy of the phage cocktail in controlling *Salmonella* on a commercial scale was not explored in this study but will be in future work.

## Figures and Tables

**Figure 1 animals-12-03087-f001:**
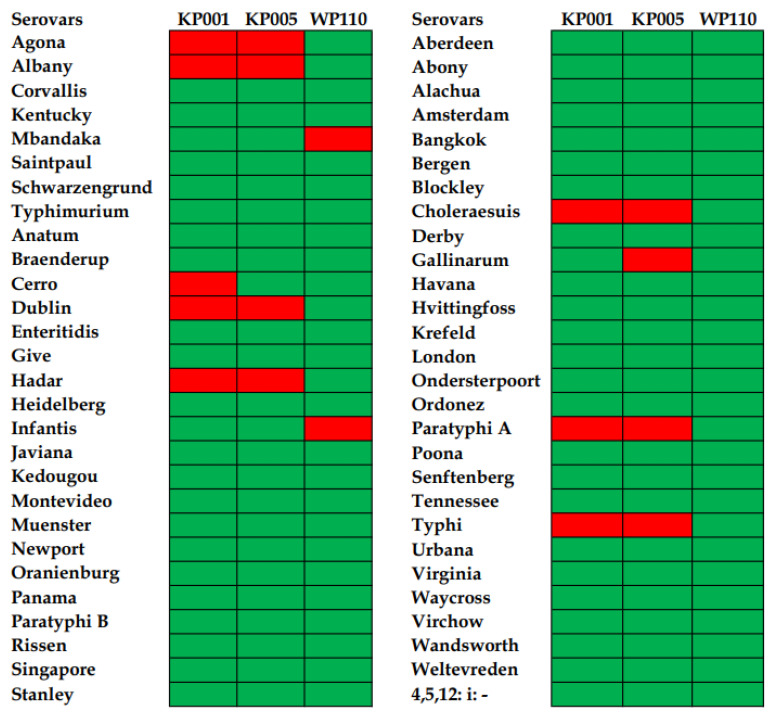
Heatmap representation of lysis profiles of three *Salmonella* phages tested on 56 *Salmonella* serovars. Green areas indicate lysis, and red areas indicate no lysis.

**Table 1 animals-12-03087-t001:** Efficacy of a phage cocktail on *Salmonella* reference serovars.

*Salmonella* Serovars	Time (h)	Bacterial Count (log CFU/mL) ^1^		
		Control	Phage Cocktail at Low MOI (10^3^)	Phage Cocktail at High MOI (10^4^)
Enteritidis	0	3.9 ± 0.0 ^a^	3.9 ± 0.0	3.8 ± 0.4
	6	7.7 ± 0.2 ^b^	ND	ND
	12	9.3 ± 0.1 ^c^	ND	ND
	18	9.5 ± 0.2 ^c^	ND	ND
Typhimurium	0	3.9 ± 0.1 ^a^	3.9 ± 0.2	4.1 ± 0.2
	6	8.0 ± 0.0 ^b^	ND	ND
	12	8.3 ± 0.1 ^b^	ND	ND
	18	9.6 ± 0.1 ^c^	ND	ND

^1^ All values provided are expressed as mean ± standard deviation in triplicate. The lowercase letters are for the control or phage treatments, and those connected by different letters indicate a significant difference (*p* < 0.05), and “ND” refers to no bacteria cells detected.

**Table 2 animals-12-03087-t002:** The number of viable *Salmonella* counts when treated with individual phages.

*Salmonella* Serovar	Time (h)	Bacterial Count (log CFU/mL) ^1^			
		Control	KP001	KP005	WP110
Enteritidis	0	3.8 ± 0.0 ^a^	3.7 ± 0.0 ^a^	3.6 ± 0.1 ^a^	3.8 ± 0.2 ^a^
	6	7.5 ± 0.2 ^b^	4.9 ± 0.2 ^b^*	5.3 ± 0.3 ^b^*	5.2 ± 0.1 ^b^*
	12	8.3 ± 0.0 ^c^	7.6 ± 0.0 ^c^*	6.5 ± 0.0 ^c^*	7.4 ± 0.3 ^c^*
	18	9.2 ± 0.1 ^d^	8.1 ± 0.1 ^d^*	7.4 ± 0.0 ^d^*	7.6 ± 0.4 ^c^*
Typhimurium	0	3.7 ± 0.3 ^a^	3.9 ± 0.3 ^a^	3.4 ± 0.3 ^a^	3.5 ± 0.1 ^a^
	6	7.9 ± 0.2 ^b^	5.2 ± 0.1 ^b^*	4.4 ± 0.1 ^b^*	5.0 ± 0.0 ^b^*
	12	8.4 ± 0.2 ^c^	5.5 ± 0.0 ^b^*	6.7 ± 0.0 ^c^*	5.2 ± 0.1 ^b^*
	18	9.3 ± 0.0 ^d^	5.9 ± 0.2 ^b^*	6.5 ± 0.5 ^c^*	6.9 ± 0.2 ^c^*

^1^ All values provided are expressed as mean ± standard deviation in triplicate. The lowercase letters are for the control or each phage treatment, and those connected by different letters indicate a significant difference (*p* < 0.05), and the *asterisks* (*) indicates a significant difference (*p* < 0.05) between the control and phage treatments at the same time.

**Table 3 animals-12-03087-t003:** Phage lysis on *Salmonella* reference serovars upon three-passage treatments of a phage cocktail.

*Salmonella*	Treatment	Lysis Ability of Cocktail ^1^ (Phage Titer log PFU/mL)											
		Passage											
		1				2				3			
		7	6	5	4	7	6	5	4	7	6	5	4
*S.* Enteritidis	Control	+++	++	++	+	++	+	+	-	++	++	-	-
	Cocktail	++	++	+	+	++	++	+	-	++	++	-	-
*S.* Typhimurium	Control	+++	+++	++	+	+++	++	+	-	+++	++	+	-
	Cocktail	+++	++	+	+	+++	+++	+	+	+++	++	+	-

^1^ Lysis ability: +++ indicates strong lysis; ++ indicates medium lysis; + indicates weak lysis; − indicates no lysis. The control is the culture without prior phage cocktail treatment.

**Table 4 animals-12-03087-t004:** Cell viability (%) of human dermal fibroblasts after treatment with a single phage.

Phage	Time (h)	Cell Viability (%)				
		Phage Concentration (log PFU/mL) ^1^				
		Control	6.0	7.0	8.0	9.0
vB_SenS_KP001	24	100	>100	>100	>100	>100
	48	100	>100	>100	>100	>100
	72	100	>100	>100	>100	>100
vB_SenS_KP005	24	100	>100	>100	>100	>100
	48	100	>100	92.4 ± 7.8 *	95.4 ± 3.4 *	>100
	72	100	>100	>100	>100	>100
vB_SenS_WP110	24	100	>100	>100	>100	>100
	48	100	>100	>100	>100	>100
	72	100	>100	>100	>100	>100

^1^ All values are provided as mean ± standard deviation of triplicates. The % cell viability >100% indicates cell proliferation over the period of study; therefore, statistical analysis was performed only on those that were <100%, and the *asterisks* (*) indicates the significant difference (*p* < 0.05) between no-phage control and treatment at the same time. The % cell viability of control was normalized at 100% for each sampling time.

**Table 5 animals-12-03087-t005:** Cell viability (%) of Caco-2 cells after treatment with a single phage.

Phage	Time (h)	Cell Viability (%) ^1^				
		Phage Concentration (log PFU/mL)				
		Control	6.0	7.0	8.0	9.0
vB_SenS_KP001	24	100	96.2 ± 4.2	97.1 ± 3.5	96.1 ± 4.4	98.9 ± 3.7
	48	100	>100	99.5 ± 1.7	98.3 ± 2.3	98.6 ± 4.4
	72	100	>100	>100	95.9 ± 3.1 *	97.6 ± 2.9 *
vB_SenS_KP005	24	100	96.7 ± 2.2	96.3 ± 1.7	97.0 ± 3.7	96.0 ± 3.8
	48	100	>100	99.2 ± 2.1	98.6 ± 1.6	96.3 ± 5.6
	72	100	>100	>100	99.3 ± 1.5	96.9 ± 2.3 *
vB_SenS_WP110	24	100	96.0 ± 2.8 *	95.9 ± 2.9 *	97.0 ± 2.7	99.1 ± 6.5
	48	100	97.7 ± 2.7 *	99.3 ± 1.9	99.8 ± 1.3	99.2 ± 2.0
	72	100	96.4 ± 1.2 *	95.7 ± 3.3 *	95.3 ± 4.2 *	98.8 ± 3.1

^1^ All values are provided as mean ± standard deviation of triplicates. The % cell viability >100% indicates cell proliferation over the period of study; therefore, statistical analysis was performed only on those that were < 100%, and the *asterisks* (*) indicates the significant difference (*p* < 0.05) between no-phage control and treatment at the same time. The % cell viability of control was normalized at 100% for each sampling time.

**Table 6 animals-12-03087-t006:** *Salmonella* prevalence in cloacal swab sample of broilers.

Time of Treatment (Days)	Age of Broiler (Days)	*Salmonella* Prevalence (Positive Samples/Total Samples) (%)		
		Control	Feeding Tube	Nipple Waterer
0	11	8/10 (80)	9/10 (90)	7/10 (70)
4	15	7/10 (70)	0/10 (0)	4/10 (40)
13	24	6/10 (60)	0/10 (0)	0/10 (0)
16	27	10/10 (100)	0/10(0)	0/10 (0)
20	31	10/10 (100)	0/10 (0)	0/10 (0)

## Data Availability

The data presented in this study are available on request from the corresponding author.

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
