# Peer review of "Oral Administration of a Phage Cocktail to Reduce *Salmonella* Colonization in Broiler Gastrointestinal Tract—A Pilot Study"

_animals, 2022, doi:10.3390/ani12223087_

Round 1
Reviewer 1 Report
This study investigated the phage survivability and the phage cytotoxic effect of a cocktail as well as individual phages for treatment and control Salmonella. The manuscript is scientifically sound and the findings are relatively explained. I recommend its acceptance for publication after minor revision.
1) L105-106, For the phages KP001, KP005 and WP110, what were the criteria for selecting.
2) The authors could expand the discussion on Salmonella phages tested on 56 serovars.
Author Response
#Reviewer 1
1) L105-106, For the phages KP001, KP005 and WP110, what were the criteria for selecting.
Answers: The ability of phage lysis is the criteria of phage selection. Three phages from our collection included here showed the highest lytic ability covering Salmonella serovars recovered from broiler production chain and other sources. Phage KP001 (formerly KP1) and KP005 (formerly KP5) were studied by Petsong et al. (2019), and phage WP110 was studied by Pelyuntha et al. (2021).
References
Pelyuntha, W., Ngasaman, R., Yingkajorn, M., Chukiatsiri, K., Benjakul, S., & Vongkamjan, K. (2021). Isolation and characterization of potential Salmonella phages targeting multidrug-resistant and major serovars of Salmonella derived from broiler production chain in Thailand. Frontiers in Microbiology, 12, 662461.
Petsong, K.; Benjakul, S.; Chaturongakul, S.; Switt, A.I.M.; Vongkamjan, K. Lysis profiles of Salmonella phages on Salmonella isolates from various sources and efficiency of a phage cocktail against S. Enteritidis and S. Typhimurium. Microorganisms. 2019, 7, 100.
2) The authors could expand the discussion on Salmonella phages tested on 56 serovars.
Answers: Discussion was added according to the reviewer’s suggestion (Line 355-359).
Reviewer 2 Report
Authors focused on the application of phage treatment to control Salmonella ib broilers. Georgia is well known center to treatment with bacteriophages. Phage therapy is a safe and effective treatment used in human medicine against Streptococcus, Staphylococcus, Enterococcus, E. coli, Shigella, Salmonella, Klebsiella, Proteus sp., Pseudomonas. The best results were noticed in acne, bronchitis, lung infections, colitis, skin infections, intestinal infections and dysbiosis. Presented article it seems to be continuation of previous work: Isolation and Characterization of Potential Salmonella Phages Targeting Multidrug-Resistant and Major Serovars of Salmonella Derived From Broiler Production Chain in Thailand (DOI: 10.3389/fmicb.2021.662461). The work is not particularly novel and does not provide useful information. Moreover, bacteriological identification of Salmonella is inaccurate, thus the quality of the work is reduced. Description of phage treatment is unclear. The methodology is chaotic and careless. Methods section needs to be revised more seriously. Moreover, it is necessary to discuss precisely and clearly, what has been discovered so far and what are the new findings. Is this important for poultry farming and industry?
Major Comments
Table S1 – lack of Kauffmann–White classification and nomenclature of serovars according to antygen O and h phases 1 and 2. Title of table is incorrect. Lack of serovars appropriate nomenclature for bacteria, that is antigenic pattern. Authors used some strain unclear codes.
Line 41 and 158 – human dermal fibroblast – lack of line name. Why human fibroblast and human epithelial cells Caco-2 wear used? Instead of that authors maight use chicken fibroblasts line.
Line 107 – 108 – Why phage isolated from sewage from a broiler farm wastewater treatment station were used? Instead of previously collected from cloacal sample from broilers farm?
Line 108 – S. Anatum A4-525 was used as a natural host… - 65 serovars of Salmonella were used, why Anatum A4-525 isolated from bovine wear used? Not serowar isolated from chicken?
Line 116 – 117 – as control of EOP used two reference strains (S. Enteritidis and S. Typhimurium) – lack of reference numer of strains and additional standard controls like reference strain E. coli
Line 129 – 130 - as control of Phage Efficacy Test In-Vitro used two human-origin strains: S. Enteritidis and S. Typhimurium S5-370. Not serovar isolated from chicken? Lack of reference numer of strains.
Line 178 Efficacy of a Phage Cocktail in Reducing Salmonella Populations in a Broiler Gastrointestinal Tract – broilers were experimentally infected with Salmonella? Tthe description is confusing. If yes, which serovar and dose (CFU, density).
Line 195 - Sampling method and sample examination - the description is confusing and chaotic. First cloacal samples were collected to 9 ml BPW next stored in an icebox (4 °C) during transportation to the laboratory for analysis. How long transport take? Normally ambient temperature guarantee stability of bacteriological swabs from 24 to 48 h. After incubation a loopful of culture fluid was further streaked directly on the XLD agar or SALMA agar? What about semi-solid agar MSRV? ON XLD we observed growth of Proteus or Citrobacter (red colony with black center), Pseudomonas. E.coli and many others when MSRV step was omitted. Moreover, some Salmonella growth without black center (H2S negative strains).
Moreover, if SALMA agar was used, BPW must be supplemented with Salmonella Supp Tab! According to manual provided by Biomérieux company. Incubation 16 to 24h at 41.5±1°C, not 37 °C for 24 h. like described in methods. Additionally, Biomérieux recommended using agglutination latex test! Due to pink colony of some strains of Enterobacteriaceae.
The prevalence of Salmonella in all samples was reported. Based on which tests? Serotyping or PCR?
If identification of Salmonella was based on questionable methods, phage cocktail might to reduce Salmonella or other similar Enterobacteriaceae.
Line 201 Their entrails, including, liver and spleen were aseptically collected for the detection of invasive Salmonella infection in their internal organs. What about intestine?
Line 209 - Biochemical tests including urease test, lysine iron sugar (LIA) test, and triple sugar iron (TSI) test were also confirmed – these tests are not sufficient to identify Salmonella spp.
Line 224 – For evaluation of a phage cocktail in vitro, only 2 human-origin strains S. Enteritidis and S. Typhimurium S5-370 were used, instead of chicken-origin. Lack of reference numer of strain.
Line 266 - as above
Line 448, 478, 495, - 3 autocitation, reference 8, 22, 31
Minor Comments
Line 95 Fifty-six in our collection were used in this study.
Line 132 - incubation with agitation (220 rpm)
Line 184 – more details about weight of broilers and feed system should be added
Line 210 - Entail samples – samples involves to something?
Line 212 - rigorously – rather vigorously
Line 226 - lysis ability against 48 of 56 serovars – results was shown as heatmap, in my opinion, unclear and indecipherable. Table will be better.
Line 308 Discussion - This suggests that this developed phage cocktail is active and survived in each part of the broiler gastrointestinal tract. Its only assumption. Lack of data from gastrointestinal tract of euthanized broilers.
Line 495 – reference 31 is not indexed in Pubmed
Major Comments
Table S1 – lack of Kauffmann–White classification and nomenclature of serovars according to antygen O and h phases 1 and 2. Title of table is incorrect. Lack of serovars appropriate nomenclature for bacteria, that is antigenic pattern. Authors used some strain unclear codes.
Line 41 and 158 – human dermal fibroblast – lack of line name. Why human fibroblast and human epithelial cells Caco-2 wear used? Instead of that authors maight use chicken fibroblasts line.
Line 107 – 108 – Why phage isolated from sewage from a broiler farm wastewater treatment station were used? Instead of previously collected from cloacal sample from broilers farm?
Line 108 – S. Anatum A4-525 was used as a natural host… - 65 serovars of Salmonella were used, why Anatum A4-525 isolated from bovine wear used? Not serowar isolated from chicken?
Line 116 – 117 – as control of EOP used two reference strains (S. Enteritidis and S. Typhimurium) – lack of reference numer of strains and additional standard controls like reference strain E. coli
Line 129 – 130 - as control of Phage Efficacy Test In-Vitro used two human-origin strains: S. Enteritidis and S. Typhimurium S5-370. Not serovar isolated from chicken? Lack of reference numer of strains.
Line 178 Efficacy of a Phage Cocktail in Reducing Salmonella Populations in a Broiler Gastrointestinal Tract – broilers were experimentally infected with Salmonella? Tthe description is confusing. If yes, which serovar and dose (CFU, density).
Line 195 - Sampling method and sample examination - the description is confusing and chaotic. First cloacal samples were collected to 9 ml BPW next stored in an icebox (4 °C) during transportation to the laboratory for analysis. How long transport take? Normally ambient temperature guarantee stability of bacteriological swabs from 24 to 48 h. After incubation a loopful of culture fluid was further streaked directly on the XLD agar or SALMA agar? What about semi-solid agar MSRV? ON XLD we observed growth of Proteus or Citrobacter (red colony with black center), Pseudomonas. E.coli and many others when MSRV step was omitted. Moreover, some Salmonella growth without black center (H2S negative strains).
Moreover, if SALMA agar was used, BPW must be supplemented with Salmonella Supp Tab! According to manual provided by Biomérieux company. Incubation 16 to 24h at 41.5±1°C, not 37 °C for 24 h. like described in methods. Additionally, Biomérieux recommended using agglutination latex test! Due to pink colony of some strains of Enterobacteriaceae.
The prevalence of Salmonella in all samples was reported. Based on which tests? Serotyping or PCR?
If identification of Salmonella was based on questionable methods, phage cocktail might to reduce Salmonella or other similar Enterobacteriaceae.
Line 201 Their entrails, including, liver and spleen were aseptically collected for the detection of invasive Salmonella infection in their internal organs. What about intestine?
Line 209 - Biochemical tests including urease test, lysine iron sugar (LIA) test, and triple sugar iron (TSI) test were also confirmed – these tests are not sufficient to identify Salmonella spp.
Line 224 – For evaluation of a phage cocktail in vitro, only 2 human-origin strains S. Enteritidis and S. Typhimurium S5-370 were used, instead of chicken-origin. Lack of reference numer of strain.
Line 266 - as above
Line 448, 478, 495, - 3 autocitation, reference 8, 22, 31
Minor Comments
Line 95 Fifty-six in our collection were used in this study.
Line 132 - incubation with agitation (220 rpm)
Line 184 – more details about weight of broilers and feed system should be added
Line 210 - Entail samples – samples involves to something?
Line 212 - rigorously – rather vigorously
Line 226 - lysis ability against 48 of 56 serovars – results was shown as heatmap, in my opinion, unclear and indecipherable. Table will be better.
Line 308 Discussion - This suggests that this developed phage cocktail is active and survived in each part of the broiler gastrointestinal tract. Its only assumption. Lack of data from gastrointestinal tract of euthanized broilers.
Line 495 – reference 31 is not indexed in Pubmed
Author Response
#Reviewer 2
Major Comments
Table S1 – lack of Kauffmann–White classification and nomenclature of serovars according to antigen O and h phases 1 and 2. Title of table is incorrect. Lack of serovars appropriate nomenclature for bacteria, that is antigenic pattern. Authors used some strain unclear codes.
Answer: In this MS, we did perform the antigenic determinants of each Salmonella serovars. Because most of serovars were received from many external sources and further kept being our collection. Some serovars from commercial broiler farm were identified and classified from our previous study while some serovars were received from culture collection bank (DMST). So, the antigenic pattern is not required for Table S1. Code strains (code names) for each serovar have already provided in the Table.
Line 41 and 158 – human dermal fibroblast – lack of line name. Why human fibroblast and human epithelial cells Caco-2 wear used? Instead of that authors might use chicken fibroblasts line.
Answer: We used primary human dermal fibroblast (HDFn) PSC-201-010TM and Caco-2 cells HTB-37™ from ATCC collection. In this study, we used human dermal fibroblast and Caco-2 cells for testing the cytotoxicity assay according to the recommendations of the animal ethic committee and specialized veterinarians in this project. They required the safety result of the phage cocktail on any mammalian cells before we accessed the phage cocktail on the broiler. Due to the strength and sensitivity of cell line along with the availability of cell lines in our laboratory collection, we decided to use the human dermal fibroblast and Caco-2 cells. We hypothesized that skin cells are the natural protective barrier to protect us from external risks including pathogens and allergens. If phages can cause toxicity to skin cells, they will subsequently make irritation and damage on the skin, and they would be unsafe. For Caco-2 cells, we performed for rearing the condition during phages entry to gastrointestinal tracts and will not cause the damage on the cells. They would be safe. There are the reasons why we chose human dermal fibroblast and Caco-2 cell in our study.
References of cell line:
https://www.atcc.org/products/pcs-201-010
https://www.atcc.org/products/htb-37
Line 107 – 108 – Why phage isolated from sewage from a broiler farm wastewater treatment station were used? Instead of previously collected from cloacal sample from broilers farm?
Answer: Phages can be isolated from many sources. For further application, phages with the highest lytic properties should be chosen as the first criteria of phage selection in stead of the source of origin.
Line 108 – S. Anatum A4-525 was used as a natural host… - 65 serovars of Salmonella were used, why Anatum A4-525 isolated from bovine were used? Not serovar isolated from chicken?
Answer: Because of S. Anatum A4-525 is the original host for phage KP001 and KP005 isolation and propagation. These phages were isolated from our previous work. If we change to another host for propagation, it might correlate with their phage reproductive cycle. That is why we still used S. Anatum A4-525 as a host for these phages.
Line 116 – 117 – as control of EOP used two reference strains (S. Enteritidis and S. Typhimurium) – lack of reference number of strains and additional standard controls like reference strain E. coli
Answer: Due to the spot test from our previous study (unpublished data), our phages could not lyse the reference E. coli strain (ATCC 25922) and other E. coli isolates in our laboratory collection. That is why we did not perform the EOP study on E. coli strains. If we perform the EOP, the result will show as ineffective.
Line 129 – 130 - as control of phage Efficacy Test In-Vitro used two human-origin strains: S. Enteritidis and S. Typhimurium S5-370. Not serovar isolated from chicken? Lack of reference number of strains.
Answer: To the best of my knowledge, Enteritidis and Typhimurium are the major serovars contaminated over broiler/poultry production chain in Thailand and several parts of the world, and subsequently cause diseases in humans and animals. We decided to choose two serovars as representatives.
References:
Ferrari, R. G., Rosario, D. K., Cunha-Neto, A., Mano, S. B., Figueiredo, E. E., & Conte-Junior, C. A. (2019). Worldwide epidemiology of Salmonella serovars in animal-based foods: a meta-analysis. Applied and Environmental Microbiology, 85(14), e00591-19.
Bangtrakulnonth, A., Pornreongwong, S., Pulsrikarn, C., Sawanpanyalert, P., Hendriksen, R. S., Wong, D. M. L. F., & Aarestrup, F. M. (2004). Salmonella serovars from humans and other sources in Thailand, 1993–2002. Emerging Infectious Diseases, 10(1), 131.
Pelyuntha, W., Ngasaman, R., Yingkajorn, M., Chukiatsiri, K., Benjakul, S., & Vongkamjan, K. (2021). Isolation and characterization of potential Salmonella phages targeting multidrug-resistant and major serovars of Salmonella derived from broiler production chain in Thailand. Frontiers in Microbiology, 12, 662461.
Afshari, A., Baratpour, A., Khanzade, S., & Jamshidi, A. (2018). Salmonella Enteritidis and Salmonella Typhimorium identification in poultry carcasses. Iranian Journal of microbiology, 10(1), 45.
Pelyuntha, W., Ngasaman, R., Yingkajorn, M., Chukiatsiri, K., & Vongkamjan, K. (2022). Decontamination of major Salmonella serovars derived from poultry farms on eggs using Salmonella phage cocktail. Asia Pacific Journal of Science and Technology, 27(2): APST-27-02-16.
Yue, M., Liu, D., Li, X., Jin, S., Hu, X., Zhao, X., & Wu, Y. (2022). Epidemiology, Serotype and Resistance of Salmonella Isolates from a Children’s Hospital in Hangzhou, Zhejiang, China, 2006–2021. Infection and Drug Resistance, 4735-4748.
Shivaning Karabasanavar, N., Benakabhat Madhavaprasad, C., Agalagandi Gopalakrishna, S., Hiremath, J., Shivanagowda Patil, G., & B Barbuddhe, S. (2020). Prevalence of Salmonella serotypes S. Enteritidis and S. Typhimurium in poultry and poultry products. Journal of Food Safety, 40(6), e12852.
Gast, R. K., Guraya, R., Jones, D. R., & Anderson, K. E. (2013). Colonization of internal organs by Salmonella Enteritidis in experimentally infected laying hens housed in conventional or enriched cages. Poultry Science, 92(2), 468-473.
Gast, R. K., Jones, D. R., Guraya, R., Anderson, K. E., & Karcher, D. M. (2020). Horizontal transmission and internal organ colonization by Salmonella Enteritidis and Salmonella Kentucky in experimentally infected laying hens in indoor cage-free housing. Poultry Science, 99(11), 6071-6074.
Line 178 Efficacy of a Phage Cocktail in Reducing Salmonella Populations in a Broiler Gastrointestinal Tract – broilers were experimentally infected with Salmonella? The description is confusing. If yes, which serovar and dose (CFU, density).
Answer: No Salmonella was inoculated into experimental broilers. We used natural occurring Salmonella present in broiler farm. We performed the baseline measurement of Salmonella prevalence before phage administered and this farm has historically reported as high prevalence of Salmonella over many cycles of broiler cultivation. The result of prevalence at baseline measurement was reported in Section 3.6 and Table 6 in revised MS.
Line 195 - Sampling method and sample examination - the description is confusing and chaotic. First cloacal samples were collected to 9 ml BPW next stored in an icebox (4 °C) during transportation to the laboratory for analysis. How long transport take? Normally ambient temperature guarantee stability of bacteriological swabs from 24 to 48 h. After incubation a loopful of culture fluid was further streaked directly on the XLD agar or SALMA agar? What about semi-solid agar MSRV? ON XLD we observed growth of Proteus or Citrobacter (red colony with black center), Pseudomonas. E. coli and many others when MSRV step was omitted. Moreover, some Salmonella growth without black center (H2S negative strains)
Moreover, if SALMA agar was used, BPW must be supplemented with Salmonella Supp Tab! According to manual provided by Biomérieux company. Incubation 16 to 24h at 41.5±1°C, not 37 °C for 24 h. like described in methods. Additionally, Biomérieux recommended using agglutination latex test! Due to pink colony of some strains of Enterobacteriaceae.
The prevalence of Salmonella in all samples was reported. Based on which tests? Serotyping or PCR?
If identification of Salmonella was based on questionable methods, phage cocktail might to reduce Salmonella or other similar Enterobacteriaceae.
Answer:
-Transport time to the laboratory is approximately 1 to 2 h.
-We kept all swab samples in an ice box at cool temperature (~4°C) during sampling and transportation. After arrival to laboratory, samples were immediately handled. So, no samples were stored at ambient temperature.
- We used XLD agar along with SALMA plate for comparison, but only Salmonella present in SALMA plate was used for further experiments. Due to its high specificity and easy observation via differentiate properties. Colonies of Salmonella will appear as pink to purple colonies while other Enterobacteriaceae will show blue or white colonies according to the manufacturer’s recommendation. MSRV was not conducted.
- We used BPW supplemented with Salmonella Supp Tab for sample enrichment. BPW (225 mL) and one tablet of Salmonella Supp Tab were mixed and roughly shake prior to aseptically distribute into sampling tube (9 mL). Additional information was added according to the reviewer’s recommendation.
- The prevalence of Salmonella was reported after serotyping. Additional information was added according to the reviewer’s recommendation.
Line 201 Their entrails, including, liver and spleen were aseptically collected for the detection of invasive Salmonella infection in their internal organs. What about intestine?
Answer: Intestine is the common organ that Salmonella can colonize. We can detect the Salmonella colonization in intestine through cloacal swab. But, in this assay, we explore the “invasive Salmonella”. Liver, caecum, and spleen are the most internal organ that Salmonella can translocate or invade. That is why we performed the assay on these organs.
References:
Silva, R. C., Cardoso, W. M., Teixeira, R. S. C., Horn, R. V., Cavalcanti, C. M., Almeida, C. P., ... & Freitas, M. L. (2015). Recovery of Salmonella Gallinarum in the organs of experimentally-inoculated Japanese quails (Coturnix coturnix). Brazilian Journal of Poultry Science, 17, 281-286.
Gast, R. K., Guraya, R., Jones, D. R., & Anderson, K. E. (2013). Colonization of internal organs by Salmonella Enteritidis in experimentally infected laying hens housed in conventional or enriched cages. Poultry Science, 92(2), 468-473.
Gast, R. K., Jones, D. R., Guraya, R., Anderson, K. E., & Karcher, D. M. (2020). Horizontal transmission and internal organ colonization by Salmonella Enteritidis and Salmonella Kentucky in experimentally infected laying hens in indoor cage-free housing. Poultry Science, 99(11), 6071-6074.
Line 209 - Biochemical tests including urease test, lysine iron sugar (LIA) test, and triple sugar iron (TSI) test were also confirmed – these tests are not sufficient to identify Salmonella spp.
Answer: Latex agglutination test was also performed along with biochemical tests. The additional detail was added into the MS.
Line 224 – For evaluation of a phage cocktail in vitro, only 2 human-origin strains S. Enteritidis and S. Typhimurium S5-370 were used, instead of chicken-origin. Lack of reference number of strain.
Line 266 - as above
Answer: To the best of my knowledge, Enteritidis and Typhimurium are the major serovars contaminated over broiler/poultry production chain in Thailand and several parts of the world, and subsequently cause diseases in humans and animals. We decided to choose two serovars as representatives.
References:
Ferrari, R. G., Rosario, D. K., Cunha-Neto, A., Mano, S. B., Figueiredo, E. E., & Conte-Junior, C. A. (2019). Worldwide epidemiology of Salmonella serovars in animal-based foods: a meta-analysis. Applied and Environmental Microbiology, 85(14), e00591-19.
Bangtrakulnonth, A., Pornreongwong, S., Pulsrikarn, C., Sawanpanyalert, P., Hendriksen, R. S., Wong, D. M. L. F., & Aarestrup, F. M. (2004). Salmonella serovars from humans and other sources in Thailand, 1993–2002. Emerging Infectious Diseases, 10(1), 131.
Pelyuntha, W., Ngasaman, R., Yingkajorn, M., Chukiatsiri, K., Benjakul, S., & Vongkamjan, K. (2021). Isolation and characterization of potential Salmonella phages targeting multidrug-resistant and major serovars of Salmonella derived from broiler production chain in Thailand. Frontiers in Microbiology, 12, 662461.
Afshari, A., Baratpour, A., Khanzade, S., & Jamshidi, A. (2018). Salmonella Enteritidis and Salmonella Typhimorium identification in poultry carcasses. Iranian Journal of Microbiology, 10(1), 45.
Pelyuntha, W., Ngasaman, R., Yingkajorn, M., Chukiatsiri, K., & Vongkamjan, K. (2022). Decontamination of major Salmonella serovars derived from poultry farms on eggs using Salmonella phage cocktail. Asia Pacific Journal of Science and Technology, 27(2): APST-27-02-16.
Yue, M., Liu, D., Li, X., Jin, S., Hu, X., Zhao, X., & Wu, Y. (2022). Epidemiology, Serotype and Resistance of Salmonella Isolates from a Children’s Hospital in Hangzhou, Zhejiang, China, 2006–2021. Infection and Drug Resistance, 4735-4748.
Shivaning Karabasanavar, N., Benakabhat Madhavaprasad, C., Agalagandi Gopalakrishna, S., Hiremath, J., Shivanagowda Patil, G., & B Barbuddhe, S. (2020). Prevalence of Salmonella serotypes S. Enteritidis and S. Typhimurium in poultry and poultry products. Journal of Food Safety, 40(6), e12852.
Gast, R. K., Guraya, R., Jones, D. R., & Anderson, K. E. (2013). Colonization of internal organs by Salmonella Enteritidis in experimentally infected laying hens housed in conventional or enriched cages. Poultry Science, 92(2), 468-473.
Gast, R. K., Jones, D. R., Guraya, R., Anderson, K. E., & Karcher, D. M. (2020). Horizontal transmission and internal organ colonization by Salmonella Enteritidis and Salmonella Kentucky in experimentally infected laying hens in indoor cage-free housing. Poultry Science, 99(11), 6071-6074.
Line 448, 478, 495, - 3 autocitation, reference 8, 22, 31
Answer: These references are very useful for explaining our methodology, result, and discussion.
Minor Comments
Line 95 Fifty-six in our collection were used in this study.
Answer: The sentence was revised according to the reviewer’s comment.
Line 132 incubation with agitation (220 rpm)
Answer: The sentence was revised according to the reviewer’s comment.
Line 184 – more details about weight of broilers and feed system should be added
Answer: Weight of broilers and feed system were added into the manuscript according to the reviewer’s suggestion.
Line 210 - Entail samples – samples involves to something?
Answer: The sentence was revised according to the reviewer’s comment.
Line 212 - rigorously – rather vigorously
Answer: The word was revised according to the reviewer’s comment.
Line 226 - lysis ability against 48 of 56 serovars – results was shown as heatmap, in my opinion, unclear and indecipherable. Table will be better.
Answer: We would like to keep the result as heatmap.
Line 308 Discussion - This suggests that this developed phage cocktail is active and survived in each part of the broiler gastrointestinal tract. Its only assumption. Lack of data from gastrointestinal tract of euthanized broilers.
Answer: The word “suggests” was changed into “assume” according to the reviewer’s recommendation.
Line 495 – reference 31 is not indexed in PubMed
Answer: Thank you for letting us know. Reference 31 is indexed in Scopus, SCImago, and Thai Citation Index.
Reviewer 3 Report
In this manuscript the authors report a preliminary study on the effectiveness of a phage cocktail in reducing Salmonella colonization of the broiler gastrointestinal tract, which has implications for controlling salmonellosis and for improving food safety.
I suggest that this manuscript could be suitable for publication in Animals, if the text is subjected to many corrections and changes, as follows:
Line 17. Insert “the” before “poultry” and insert a comma after “concern”
Line 18. Change “approach is” to “measures are”
Line 20. Change “the” to “a”
Line 22. Change “safety in which” to “safety, whereby”
Line 24. Remove “our”
Line 25. Insert “an” before “effective”
Line 26. Insert “the” before “gastrointestinal” and change “in broilers” to “of broilers”
Line 30. Change “on” to “to”
Line 31. Insert “a” before “broiler”
Line 33. Insert “a” before “broiler”
Line 34. Change “could reduce S. Enteritidis and S. Typhimurium” to “reduced Salmonella enterica serovars Enteritidis and Typhimurium”
Line 36. Change “A developed” to “The” and remove “a”
Line 37. Change “3” to “three”
Line 40. Change “phage comprised in a phage cocktail at a concentration up” to “phage was in the phage cocktail at a concentration of up” and insert a full-stop after “PFU/mL”
Line 41. Insert “These” before “did” and change “and” to “or”
Line 42. Insert a comma after “viability” and change “of the co-culture” to “of co-culture”
Line 43. Change “5 doses” to “five doses”
Line 45. Remove “in”
Line 46. Change “Findings” to “Our findings”
Line 47. Insert “agent” after “biocontrol” and insert a comma after “broilers”
Line 56. “S. Enteritidis and S. Typhimurium are the most serovars of” to “Salmonella enterica serovars Enteritidis and Typhimurium are the most common causes of”
Line 61. Change “Therefore, control of Salmonella in the primary” to “Control of Salmonella in poultry primary”
Line 66. Insert a comma after “common”
Line 67. Insert a comma after “antibiotics
Line 68. Insert “a” before “broiler” and change “consecutive” to “the subsequent”
Line 73. Insert “status” before “by”
Line 77. Italicise “Salmonella” and change “successful” to “successfully”
Line 82. Insert “an” before “undetectable”
Line 83. Change “the 34” to “a 34”
Line 94. Change “Condition” to “Conditions”
Line 95. Italicise “Salmonella”
Line 112. Insert a full-stop after “31]”
Line 115. Change “the protocol of” to “a previously published protocol”
Line 122. Change “Efficiency of plating” to “EOP”
Line 130. Change “Bacterial” to “A bacterial”
Line 135. Insert “A” before “Salmonella”
Line 140. Change “upon” to “Upon”
Line 141. Change “treated to “treatment”
Line 142. Change “Culture” to “A culture”
Line 144. Change “the” to “a”
Line 146. Insert “an” before “additional”
Lines 146-147. Change “the protocols of [8] and [29]” to “previously published protocols [8,29]”
Line 158. Change “skin cell” to “skin cells”
Line 164. Change “cell” to “cells,”
Line 185. Change “3 groups (10 chickens” to “three groups (ten chickens”
Line 187. Change “group” to “groups”
Line 195. Change “method and sample examination” to “Method and Sample Examination”
Line 203. Italicise “Salmonella”
Line 206. Change “the” to “an”
Line 207. Insert “the” before “Biomérieux”
Line 210. Change “Entail” to “Entrail”
Line 214. Italicise “Salmonella”
Line 226. Change “Phage” to “Phages”
Line 237. Change “of EOP assay revealed that KP001” to “of the EOP assay revealed that the KP001”
Line 242. Insert “the” before “three”
Line 259. Change “bacteria cell” to “bacterial cells”
Line 285. Change “3” to “three”
Line 288. Change “survivability in simulated chicken gastrointestinal conditions” to “Survivability in Simulated Chicken Gastrointestinal Conditions”
Line 296. Change “Table” to “Tables”
Line 299. Change “triplicate” to “triplicates”
Line 300. Insert “the” before “period” and insert a comma after “study”
Line 301. Change “indicate” to “indicates”
Line 310. Change “triplicate” to “triplicates”
Line 311. Insert “the” before “period” and insert a comma after “study”
Line 312. Change “indicate” to “indicates”
Line 318. Change “the” to “a”
Line 322. Change “the” to “a”
Line 323. Remove “be”
Line 325. Change “the prevalence of at” to “a prevalence of”
Line 348. Remove “as”
Line 349. Change “than” to “over”
Line 365. Change “the chance of a phage” to “the possibility of phage”
Line 367. Change “composed” to “present”
Line 369. Insert “a” before “farm”
Line 373. Change “other” to “another”
Line 379. Change “stimulated” to “simulated”
Line 380. Change “this developed” to “our developed”
Line 382. Change “ion” to “ions”
Line 385. Insert “a” before “cocktail”
Line 388. Change “reveals” to “revealed”
Line 392. Change “were” to “was”
Line 395. Insert “presented here” after “data”
Line 397. Change “day” to “days”
Line 398. Change “greater than” to “preferable to a”
Line 400. Change “phages” to “phage” and insert “a” before “nipple”
Line 403. Change “in a commercial condition” to “under commercial conditions”
Line 404. Insert a comma after “guts”
Line 413. Change “not conducted in this study which will be consequently conducted in future work.” to “not explored in this study, but will be in future work.”
Author Response
#Reviewer 3
Line 17. Insert “the” before “poultry” and insert a comma after “concern”
Line 18. Change “approach is” to “measures are”
Line 20. Change “the” to “a”
Line 22. Change “safety in which” to “safety, whereby”
Line 24. Remove “our”
Line 25. Insert “an” before “effective”
Line 26. Insert “the” before “gastrointestinal” and change “in broilers” to “of broilers”
Line 30. Change “on” to “to”
Line 31. Insert “a” before “broiler”
Line 33. Insert “a” before “broiler”
Line 34. Change “could reduce S. Enteritidis and S. Typhimurium” to “reduced Salmonella enterica serovars Enteritidis and Typhimurium”
Line 36. Change “A developed” to “The” and remove “a”
Line 37. Change “3” to “three”
Line 40. Change “phage comprised in a phage cocktail at a concentration up” to “phage was in the phage cocktail at a concentration of up” and insert a full-stop after “PFU/mL”
Line 41. Insert “These” before “did” and change “and” to “or”
Line 42. Insert a comma after “viability” and change “of the co-culture” to “of co-culture”
Line 43. Change “5 doses” to “five doses”
Line 45. Remove “in”
Line 46. Change “Findings” to “Our findings”
Line 47. Insert “agent” after “biocontrol” and insert a comma after “broilers”
Line 56. “S. Enteritidis and S. Typhimurium are the most serovars of” to “Salmonella enterica serovars Enteritidis and Typhimurium are the most common causes of”
Line 61. Change “Therefore, control of Salmonella in the primary” to “Control of Salmonella in poultry primary”
Line 66. Insert a comma after “common”
Line 67. Insert a comma after “antibiotics
Line 68. Insert “a” before “broiler” and change “consecutive” to “the subsequent”
Line 73. Insert “status” before “by”
Line 77. Italicise “Salmonella” and change “successful” to “successfully”
Line 82. Insert “an” before “undetectable”
Line 83. Change “the 34” to “a 34”
Line 94. Change “Condition” to “Conditions”
Line 95. Italicise “Salmonella”
Line 112. Insert a full-stop after “31]”
Line 115. Change “the protocol of” to “a previously published protocol”
Line 122. Change “Efficiency of plating” to “EOP”
Line 130. Change “Bacterial” to “A bacterial”
Line 135. Insert “A” before “Salmonella”
Line 140. Change “upon” to “Upon”
Line 141. Change “treated to “treatment”
Line 142. Change “Culture” to “A culture”
Line 144. Change “the” to “a”
Line 146. Insert “an” before “additional”
Lines 146-147. Change “the protocols of [8] and [29]” to “previously published protocols [8,29]”
Line 158. Change “skin cell” to “skin cells”
Line 164. Change “cell” to “cells,”
Line 185. Change “3 groups (10 chickens” to “three groups (ten chickens”
Line 187. Change “group” to “groups”
Line 195. Change “method and sample examination” to “Method and Sample Examination”
Line 203. Italicise “Salmonella”
Line 206. Change “the” to “an”
Line 207. Insert “the” before “Biomérieux”
Line 210. Change “Entail” to “Entrail”
Line 214. Italicise “Salmonella”
Line 226. Change “Phage” to “Phages”
Line 237. Change “of EOP assay revealed that KP001” to “of the EOP assay revealed that the KP001”
Line 242. Insert “the” before “three”
Line 259. Change “bacteria cell” to “bacterial cells”
Line 285. Change “3” to “three”
Line 288. Change “survivability in simulated chicken gastrointestinal conditions” to “Survivability in Simulated Chicken Gastrointestinal Conditions”
Line 296. Change “Table” to “Tables”
Line 299. Change “triplicate” to “triplicates”
Line 300. Insert “the” before “period” and insert a comma after “study”
Line 301. Change “indicate” to “indicates”
Line 310. Change “triplicate” to “triplicates”
Line 311. Insert “the” before “period” and insert a comma after “study”
Line 312. Change “indicate” to “indicates”
Line 318. Change “the” to “a”
Line 322. Change “the” to “a”
Line 323. Remove “be”
Line 325. Change “the prevalence of at” to “a prevalence of”
Line 348. Remove “as”
Line 349. Change “than” to “over”
Line 365. Change “the chance of a phage” to “the possibility of phage”
Line 367. Change “composed” to “present”
Line 369. Insert “a” before “farm”
Line 373. Change “other” to “another”
Line 379. Change “stimulated” to “simulated”
Line 380. Change “this developed” to “our developed”
Line 382. Change “ion” to “ions”
Line 385. Insert “a” before “cocktail”
Line 388. Change “reveals” to “revealed”
Line 392. Change “were” to “was”
Line 395. Insert “presented here” after “data”
Line 397. Change “day” to “days”
Line 398. Change “greater than” to “preferable to a”
Line 400. Change “phages” to “phage” and insert “a” before “nipple”
Line 403. Change “in a commercial condition” to “under commercial conditions”
Line 404. Insert a comma after “guts”
Line 413. Change “not conducted in this study which will be consequently conducted in future work.” to “not explored in this study, but will be in future work.”
Answer: All texts were subjected to revision according to the reviewer’s comments (Revised MS file).
Round 2
Reviewer 2 Report
Authors in replaydecribe perform the antigenic determinants of each Salmonella serovars. In my opinion it`s not true. I don`t understand the reasoning for not adding the Salmonella nomenclature of serovars according to antigen O and h phases 1 and, that would have been something easy and would have added value to the report. I do not agree with this approach. The antigenic pattern is require. While the Newport, Mbandaka, Paratyphi B serotypes belong to well-known groups, Krefeld, Aberdeen, Alachua, Bangkok, Bergen, Hvittingfoss, Urbana or Waycross and others are not.
According literature serotype is crucial for appropriate matching, especially in case of O antygen with _ presence of the somatic O antigen conditioned by the presence of phage (lysogenic conversion) e.g. Agona 1,4,12. Furthermore, flagellar antygen in [ ], which may or may not be present due to reasons other than lysogenic conversion, e.g. Agona [1,2]. Flagellar antigens in [ ] may exceptionally appear in some wild-type strains of a given serovar. The same for:
Somatic antigen 1,4,[5],12 (group O:4):
· Saintpaul, Paratyphi B, Derby
Somatic antigen 1,4,[5],12,[27] (group O:4):
· Stanley, Abony
Somatic antigen 1,4,12,27 (group O:4):
· Schwarzengrund
Somatic antigen 6,7,14 (group O:7):
· Mbandaka, Infantis, Montevideo, Oranienburg, Rissen, Tennessee
Somatic antigen 8,20 (group O:8):
· Alban, Corvallis, Kentucky, Newport,
Somatic antigen 3,10,[15] (group O:3,10): lisogenic with phage Ɛ15
· London
Somatic antigen 3,10,[15] [15,34] (group O:3,10): lisogenic with phage Ɛ15 and than Ɛ34
· Anatum, Give, Muenster, Amsterdam
Somatic antigen 1,9,12[Vi] (group O:9):
· Dublin
Somatic antigen 1,9,12 (group O:9):
· Enteritidis, Javiana, Panama, Gallinarum (Pullorum),
Somatic antigen 1,13,23 (group O:13):
· Kedougou, Havana, Ordonez, Poona
Flagellar antigen h1 or h2:
· Corvallis [z6]
· Montevideo g,m,[p],s
· Give [d],l,v
· Derby [1,2]
· Havana f,g,[s]
· Senftenberg g,[s],t
· Tennessee [1,2,7]
· Waycross [e,n,z15]
Moreover:
Did you check: Typhimurium or monophasic Typhimurium?
Did you check: Istambul O:8 or Hadar O:6,8? Phase h1 and h2 are the same.
Did you check: O:6,8 Blockley or Haardt? Phase h1 and h2 are the same.
Did you check: Choleraesuis O:6,7 ora Paratyphi C O:6,7,[Vi]? Phase h1 and h2 are the same.
Did you check: Virginia O:8 or Muenchen O:6,8? Phase h1 and h2 are the same.
Did you check presence of Vi antygen inserovar Typhi? If yes, please describe the methods.
Did you check: Poona O:13 1,13,22 or Farmsen 13,22? Phase h1 and h2 are the same. If yes, please describe the methods.
Did you check by PCR 4,5,12: i: - is monophasic Typhimurium?
Accuracy is not the strengh of this paper..
Table 1S, BAD NOMENCLATURE. Authors don`t know the name of serovar?!
Onsterport – doesn`t exist!!!
Corvallis instead of Corvalis.
Braenderup instead of Brenderup
Kedougou instead of Kedougon
Senftenberg instead of Seftenberg
If you I perform the serotyping of each Salmonella why don't want to show in the table? In my opinion due to lack of serotyping, which proves the careless preparation of the study or even lack of serotyping.
In my opinion antigenic pattern is required for Table S1
Line 158: it should ba addead: primary neonatal dermal fibroblast cells (PCS-201-010) – from ATCC collection? For cell culture Authors use , 20% (v/v) fetal bovine serum?! For primary neonatal dermal fibroblast cells usually serum-free medium or2% fetal bovine serum is recommended. Moreover Fibroblast Basal Medium with rh FGFβ, L-glutamine, Ascorbic acid, Hydrocortisone Hemisuccinate, rh Insulin are recommended.
Moreover, for cell culture primary neonatal dermal fibroblast cells (Evaluation of Phage Cytotoxicity in Cell Lines) unsuitable condition were used. Primary neonatal dermal fibroblasts needs 2% FBS and growth factors.
Line 164: Caco-2 are epithelial cells isolated from colon. For culture Eagle's Minimum Essential Medium or Dulbecco's Modified Eagle Medium/Nutrient Mixture F-12, supplemented with 1% penicillin, 20% fetal bovine serum, and L-glutamine were used. Not for HDFn.
Line 107 – 108 The best choose are phages matching to Salmonella isolated from broilers
Line 108 – S. Anatum A4-525 was used as a natural host… - 65 serovars of Salmonella were used, why Anatum A4-525 isolated from bovine were used? Not serovar isolated from chicken?
Line 107 – 108 I understand, thus title should be changed. Authors used S. Anatum isolated from bovine, thus where is the association with in broiler gastrointestinal tract? The better will be phages matching to 1 from 56 Salmonella isolated from broilers
Line 116 – 117 – as control of EOP used two reference strains (S. Enteritidis and S. Typhimurium) – lack of reference number of strains and additional standard controls like reference strain E. coli. Authors have unpublished data about resistance to phage lysis of E. coli strain (ATCC 25922) and other E. coli isolates. It should be added to methods and results.
Author not added reference number of strains S. Enteritidis and S. Typhimurium (control of EOP)
Line 129 – 130 - Authors indicate that ,Enteritidis and Typhimurium are the major serovars contaminated over broiler/poultry production chain in … several parts of the world. It`s not true. S. Enteritidis is major serowar contaminated meat production, but S. Typhimurium is rare, The most popular are serovars Infantis, Newport, Kentucky. This serovars are more resistant to antibiotics and disinfection. Morover S. Enteritidis isolated from human origin in many cases differ from isolates from broilers.
Author not added reference number of strains S. Enteritidis and S. Typhimurium S. Enteritidis S5-371 and 129 S. Typhimurium S5-370.
References from 2004, 2013, 2018, 2019 do not reflect current epidemiology of Salmonella.
Line 178 Efficacy of a Phage Cocktail in Reducing Salmonella Populations in a Broiler Gastrointestinal Tract. The description is still confusing. The farm was contaminated, but which serotype? Bacteriophage therapy is tailored therapy. Authors used S. Anatum A4-525 phage in flock/broilers infected with UNKNOWN Salmonella serotype. Serotype it should be added.
If Authors serotyping e.g. exotic Aberdeen O:11, Alachua O:35, Bangkok O:38, Bergen O:47, Hvittingfoss O:16, Urbana O:30 or Waycross O:41, must dispose broad range os serum.
I suggested that samlpes to SALMA agar must be supplemented with Salmonella Supp Tab. Authors add this adnotation, but suplement should be used immediately before incubation BPW. Authors describe adding supplements and stored in an ice 4 °C. Normally ambient temperature guarantee stability of bacteriological swabs from 24 to 48 h.
The prevalence of Salmonella was reported after serotyping. Authors perform latex agglutination test in a commercial service company. Where is the result of serotyping? Still identification of Salmonella was based on questionable methods. Due to pink colony of some strains of Enterobacteriaceae are pink SALM agar! Thus phage cocktail might to reduce Salmonella or other similar Enterobacteriaceae. Simple PCR test to presence of invA gene or use polyvalent HM serum will be appropriate to confirmation of Salmonella.
Line 206 OK
Line 213 – Authors added only information about serotyping by a commercial service company. Still these biochemical tests are not sufficient to identify Salmonella spp.
Line 253 – Authors indicate that, Enteritidis and Typhimurium are the major serovars contaminated over broiler/poultry production chain in … several parts of the world. It`s not true. S. Enteritidis is major serowar contaminated meat production, but S. Typhimurium is rare, The most popular are serovars Infantis, Newport, Kentucky. This serovars are more resistant to antibiotics and disinfection. Morover S. Enteritidis isolated from human origin in many cases differ from isolates from broilers.
Author not added reference number of strains S. Enteritidis and S. Typhimurium S. Enteritidis S5-371 and 129 S. Typhimurium S5-370.
References from 2004, 2013, 2018, 2019 do not reflect current epidemiology of Salmonella.
Line 448, 478, 495, - 3 autocitation, reference 8, 22, 31. Authors answer that references are very useful for explaining our methodology, result, and discussion, but are still self-citation.
Minor Comments
Line 95 OK
Line 132 OK
Line 184 – OK
Line 210 - OK
Line 218 - vigorously
Line 240 - heatmap is indecipherable and unclear. I suggest results in table.
Line 308 Discussion - Lack of data from gastrointestinal tract of euthanized broilers.
Line 495 – reference 31 is not indexed in PubMed
References should be described as follows, depending on the type of work:
Journal Articles:
1. Author 1, A.B.; Author 2, C.D. Title of the article. Abbreviated Journal Name Year, Volume, page range.
All references should be change, especially Volume should be italic
Reference 16, 19: change to apprioprate notation withought ( ) - 2020, 9(9): 594.
In conclusion. Major comments were not taken into account and not corrected. The basis of the research, correct and reliable identification of the Salmonella is omitted.
Article has serious flaws, additional experiments needed (serotyping, biochemical identification of Salmonella), research not conducted correctly.
Authors added some information from a very similar and cited article (self-citation, reference 8): Isolation and Characterization of Potential Salmonella Phages Targeting Multidrug-Resistant and Major Serovars of Salmonella Derived From Broiler Production Chain in Thailand.

Author Response
Dear Reviewer,
Please see the attachment. We have responded to your comments.
Thank you for your valuable suggestions.
Sincerely,
